# Production of forest seedlings using sewage sludge and automated irrigation with ozonated cattle wastewater

**Laiz de Oliveira Silva**[1�], **Henrique Vieira Mendonça**[2‡], **Bruno Antonio Augusto Faria Conforto**[3‡], **Marinaldo Ferreira Pinto**[2‡], **Daniel Fonseca de Carvalho**[2055]*

**1** Postgraduate Program in Agronomy-Soil Sciences, Institute of Agronomy, Federal Rural University of Rio de Janeiro, Seropédica, Rio de Janeiro, Brazil, **2** Department of Engineering, Institute of Technology, Federal Rural University of Rio de Janeiro, Seropédica, Rio de Janeiro, Brazil, **3** Undergraduate course in Agronomy, Institute of Agronomy, Federal Rural University of Rio de Janeiro, Seropédica, Rio de Janeiro, Brazil

055 These authors contributed equally to this work.
‡ HVM, BAAFC and MFP also contributed equally to this work.
* daniel.fonseca.carvalho@gmail.com, carvalho@ufrrj.br

**Data Availability Statement:** All relevant data are within the paper and its Supporting Information files.

## Abstract

The large volume of effluents generated by intensive cattle production can become an environmental problem, requiring solutions that combine treatment and disposal of reuse water. The quality of cattle wastewater (CWW) treated by ozonation, the water requirement and its effect on the growth of seedlings of *Dalbergia nigra* cultivated with sewage sludge were determined under different light conditions. The study was carried out in a split plot scheme with 2 shading levels (0%—$C_1$, and 49.4% attenuation—$C_2$) and 3 types of irrigation water (control–$T_1$, 1 h ozonation–$T_2$, and 2 h–$T_3$), with 4 repetitions. Direct sowing was realized into 280 $cm^3$ tubes which were irrigated by drip irrigation with automatic management. The height and collar diameter were measured every 21 days, and at the end of the nursery phase, and the Dickson quality index (DQI) and irrigation water productivity (WPir) were determined. In addition, seedlings were transplanted in a forest restauration area (FRA) of Atlantic Forest, with height and diameter monitoring for 200 days. With ozonation, there was an increase in pH and a reduction in electrical conductivity, total solids and turbidity in the CWW, allowing its use for irrigation of forest seedlings. The maximum volumes of water applied were 2.096 and 1.921 L plant$^{-1}$, with water supply $T_2$ and $T_1$, respectively, and coverages $C_1$ and $C_2$. In these conditions, the seedlings reached DQI of 0.47 and 0.17, and WPir of 2.35 and 1.48 g L$^{-1}$, respectively. The initial vegetative growth of the seedlings planted in the FRA was benefited by the nutrients provided by the CWW treated. Therefore, the use of sewage sludge and CWW treated has the potential to produce forest seedlings, reducing the release of waste and effluents into the environment.

## Introduction

Brazil has the second largest cattle herd in the world, with approximately 264 million head [1]. Despite the economic benefits for the country, contributing with 8.6% of the GDP, intensive

**Funding:** This research was funded by Fundação de Amparo à Pesquisa do Estado do Rio de Janeiro (FAPERJ, http://www.faperj.br/, grant number E-26/202.909/2018) (DFC), Conselho Nacional de Desenvolvimento Científico e Tecnológico (CNPq, http://www.cnpq.br/, grant number 310604/2018-4) (DFC) and also by Coordenação de Aperfeiçoamento de Pessoal de Nível Superior (Finance Code 001) (LOS).

**Competing interests:** The authors have declared that no competing interests exist.

cattle farming generates large volumes of effluents, which can reach 130 L animal$^{-1}$ day$^{-1}$ [2], considering the volume of feces, urine and water used to clean the pens.

Cattle wastewater (CWW) is composed of a mixture of animal waste, drinking water that has been wasted and water used to clean the confinement environment and its annexes. CWW has BOD$_5$ between 2000 and 30,000 mg L$^{-1}$ and total nitrogen ranging from 200 to 2,055 mg L$^{-1}$ [3], alarming concentrations with high polluting potential that can cause oxygen depletion and eutrophication in surface water resources [4]. Thus, disposing of the effluent generated, from its collection, transportation, storage, treatment and use, constitutes an environmental, social and economic challenge. In Brazil, the final disposal of effluents is regulated by Resolutions 357/05 and 430/11 [5, 6] of the National Environment Council (CONAMA).

In addition to management actions that minimize water consumption and effluent production in the dairy chain [7], it is necessary to treat the waste generated for its use or even disposal into water bodies. However, the cost of effluent treatment is a restrictive factor [8], causing the search for more economical and efficient technologies and processes to become a growing interest among researchers [9].

The anaerobic digestion process has been an alternative as a primary treatment of CWW. Evaluated the efficiency of the UASB (Upflow Anaerobic Sludge Blanket) reactor in removing organic pollutants (BOD$_{5.20}$, COD), total solids, volatile solids and nutrients in CWW, indicating its potential use as a biofertilizer in agriculture [10]. However, the high concentrations of nutrients, especially nitrogen and phosphate compounds, do not allow their continuous use in plant irrigation. Thus, advanced oxidation processes, as Fenton's reagent, hydrogen peroxide, ozone and photocatalyst can be employed as further treatment for this purpose, as they are capable of removing recalcitrant organic compounds in wastewater [11].

Considered a green technology [12], the use of ozone in secondary treatment of CWW has become popular in the recent years due to the decrease in its production cost in the last decade [11]. In addition, ozone does not remain solubilized for long in water, does not form toxic by-products in most cases, and is generated in situ, eliminating the need for storage or use of chemicals [13]. The O$_3$ generation is based on the corona discharge process [14], which can improve the biodegradability of wastewater [15], removing color and organic compounds and allowing the breakdown and oxidation of organic fractions that are difficult to degrade by biological mechanisms [16–18].

The reaction with organic compounds can occur directly (molecular ozone) or indirectly through the formation of secondary oxidants such as hydroxyl radicals [19], according to Eqs 1 and 2 [20, 21]:

$$O_3 + OH^- \rightarrow O_2 + HO_2^- \tag{1}$$

$$HO^\bullet + O_3 \rightarrow HO_2^\bullet + O_2 \tag{2}$$

In addition to treated wastewater, the application of sewage sludge has been carried out in forest areas aiming to enhance both tree growth and wood production and to improve several soil characteristics [21]. Called biosolid, after stabilization process, this solid waste comes from sewage treatment plants has become popular in Brazil [22–24], and used in the composition of organic substrates for seedling production [22–26]. This is a promising initiative for the forestry sector in the country [27], which has a nursery capacity of around 150 million seedlings of native forest species [28].

The use of wastewater and biosolid in agriculture is an alternative for plant development, minimizing the removal of water from natural bodies and reducing the release of effluents into the environment [29], besides avoiding the disposal of a residue with high concentrations of

organic matter and heavy metals. Associated with the use of biosolid as substrate, wastewater should be evaluated in systems for production of seedlings of forest species, which are characterized as a determining step in the regeneration of degraded areas, especially in Brazil, which has millions of hectares under this condition. However, regardless of the quality, the water supply must be done with discretion, especially in forest nurseries, which traditionally do not use irrigation management, compromising the efficiency of the system and the quality of the seedlings [24].

Among the native species of the Atlantic Forest, one of the most degraded biomes in Brazil [23], *Dalbergia nigra* (Jacarandá-da-bahia) stands out, which has been the target of inadequate exploitation due to its natural durability and high market value [30], being considered as at high risk of extinction [31]. The species belongs to the successional group of late secondary species [32], giving it better growth in intermediate levels of shading.

This study aimed to evaluate the quality of CWW treated by ozonation and its effect on water requirement and growth of *Dalbergia nigra* seedlings, using substrate composed of treated sewage sludge. Plant development in nursery phase was evaluated in full sun and under shading condition, by measuring the main morphological characteristics and chemical parameters selected in the roots and shoots. In addition, height and diameter measurements of seedlings planted in a forest restoration area were performed. A descriptive and inferential statistical analysis was performed to explain the results obtained.

## Material and methods

### Experimental area and planting of seedlings

The experiment in nursery phase was conducted from June to September 2021 in an experimental area located in the Institute of Technology of the Federal Rural University of Rio de Janeiro (coordinates: 22°45′21″S, 43°40′28″W), Brazil. The climate of the region is type Aw, according to Köppen's classification, with annual average precipitation and temperature ranging from 1,300 to 1,600 mm and from 22 to 24°C, respectively [33].

*Dalbergia nigra* (Jacarandá-da-bahia) seeds were acquired from the Brazilian Forest Institute (IBF, Curitiba-PR), with guarantees of viability and good phytosanitary conditions. Sowing was performed on 04/26/2021, with two seeds per tube (280 cm$^3$), filled with pure biosolid (sewage sludge). This material originates from a sewage treatment plant of the Rio de Janeiro State Water and Sewerage Company (*Companhia Estadual de Águas e Esgotos do Rio de Janeiro*–CEDAE), from domestic and commercial urban areas, containing no industrial waste. The chemical characterization performed according to the official procedures contained in CONAMA Resolution 375 [34] indicated total macronutrient values of 1.5% N, 0.61% P, 0.16% K, 0.89% S, 1.57% Ca and 0.32% Mg, in addition to 9.41% organic carbon. For micronutrients, the values were 722.7 ppm Zn, 20773.3 ppm Fe, 184.5 ppm Mn, 164.3 ppm Cu and 12.1 ppm B. Physical analyzes performed by [26] indicated total porosity of 0.70 cm cm$^{-3}$, mean particle density of 1.71 g cm$^{-3}$ and apparent density of 0.74 g cm$^{-3}$, compatible with the physical characteristics of humic soils.

After seedling emergence, thinning was performed, always leaving the tallest and most centralized plant. At thirty-five days and with average height of the seedlings around 7.5 cm, the tubes were placed in plastic trays (maximum capacity of 54 tubes) spaced apart with alternating seedlings, totaling 24 plants per tray (3 rows with 8 plants per row).

The benches were covered on top and on the sides by commercial, black-colored polyolefin agricultural nets (Sombrite$^{®}$), with nominal shading of 50%, except the benches that remained in full sun. In order to better characterize the treatments, the illuminance of the covers was evaluated for 5 days, at intervals of 1 hour, using a digital luxmeter (LD-400 model,

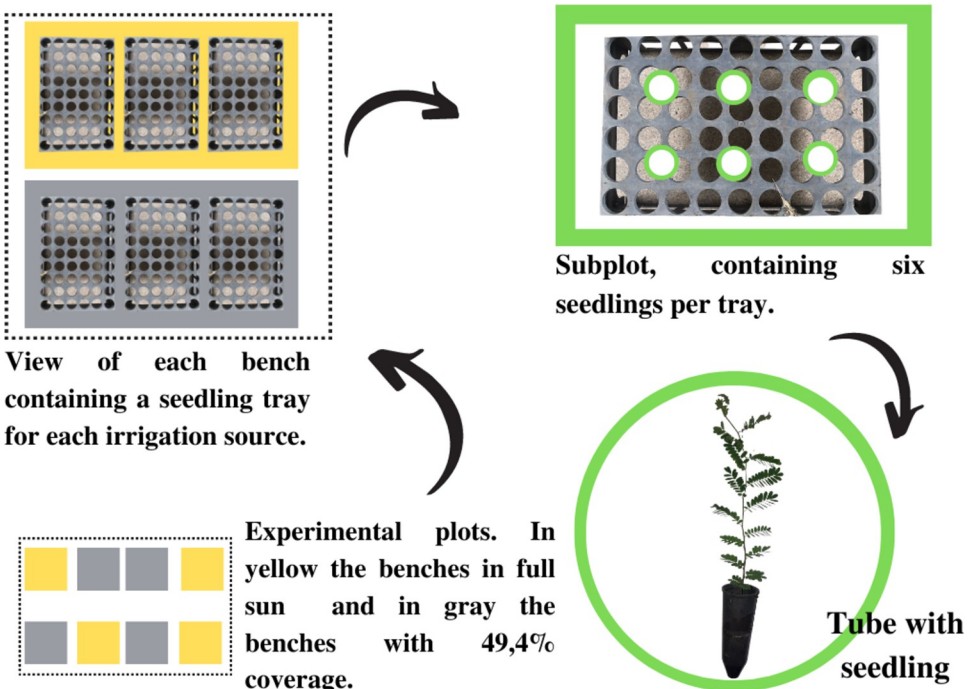

**Fig 1. Experimental scheme with two covers (plot) and three types of water for irrigation (subplot).**

Instrutherm®), as well as solar radiation and photosynthetically active radiation, respectively using a pyranometer (MP-200 model, Apogee®) and a Quantum meter (MQ-200 model, Apogee®). The mean daily illuminance values were 67.74 Lux ($C_1$) and 34.28 Lux ($C_2$), characterizing the treatments as 0 (full sun) and 49.4% attenuation, respectively.

The study was conducted in a split plot scheme with 2 levels of cover (plot) and 3 types of water for irrigation (subplot), with 4 replicates (Fig 1). Treatments consisted of local-supply water ($T_1$), considered as control, and CWW treated by ozonation for 1 h ($T_2$) and 2 h ($T_3$), with independent pumping system. Each experimental plot was composed of 6 plants, totaling 48 plants per type of water applied.

### Collection and treatment of cattle wastewater (CWW)

CWW was collected in the experimental unit called Integrated Agroecological Production System (coordinates: 22°45'7"S 43°40'14"W after preliminary treatment to remove coarse solids (decanter) and anaerobic biological treatment (UASB reactor), as described by [4]. The cattle wastewater after UASB reactor showed the following physical-chemical characteristics: pH—7.4, electrical conductivity—2.49 dS m$^{-1}$, total solids—969.8 mg L$^{-1}$, turbidity—231 NTU, total organic carbon—30.9 mg L$^{-1}$, calcium—17.8 mg L$^{-1}$, total phosphorus—17.4 mg L$^{-1}$, magnesium—13.2 mg L$^{-1}$, ammoniacal nitrogen—48.36 mg L$^{-1}$ and sodium—33.6 mg L$^{-1}$.

The treatment of CWW by ozonation was performed according to a methodology adapted by [12], on a bench scale, using fixed-bed reactors constructed with 3-inch-diameter, 0.7-m-long PVC pipe for sewage, filled with porous plastic medium (bio-rings), in 60% of its volume. The oxidation process was performed by a diffuser (20 μm pore size) arranged at the bottom of the reactor and connected by a hose to an $O_3$ generator (Ozone Generation, GL-3189A, China), with a flow rate of 1.3 L min$^{-1}$, purity of 92% (± 2) and average concentration of 7.8 mg L$^{-1}$ (±1). A piece of cotton waste rag is inserted into the reactor head to generate an internal

atmosphere of higher concentration of ozone, avoiding direct contact of atmospheric air with the surface of the gas/liquid, and to absorb particles mobilized by the force of gas drag (flotation process).

After treatment by ozonation, the following parameters were evaluated: pH, electrical conductivity (EC), total solids (TS), turbidity (NTU), total organic carbon (TOC), ammoniacal nitrogen ($NH_4^+$), total phosphorus ($P_t$) and total concentrations of sodium, calcium and magnesium ($Na^+$, $Ca^{2+}$, $Mg^{2+}$) concentrations. The analyses were performed according to standard methods [35].

Sodium adsorption ratio (SAR) was calculated using Eq(3):

$$SAR = \frac{Na^+}{\sqrt{\frac{Ca^{++}+Mg^{++}}{2}}}$$ (3)

for $Na^+$, $Ca^{++}$ and $Mg^{++}$ in *millimoles* $L^{-1}$.

## Irrigation system

The seedlings were irrigated by a drip system, using emitters (Netafim, PCJ-HCNL model) with flow rate of 3.0 L $h^{-1}$. Uniformity tests were performed for each type of water, resulting in distribution uniformity coefficient (DUC) greater than 95%. Water replacement was performed automatically, in response to the water need of the plants. This was achieved by using the simplified irrigation controller (SIC), proposed by [36], which operates in response to soil/substrate water tension and is regulated by the level difference between a ceramic micro-cup (sensor) and a pressure switch. This device has been used in irrigation management of different crops [37, 38] and in the production of seedlings of forest species [23, 24].

Six independent controllers were used, one for each treatment, whose sensors were installed vertically in the tube filled with biosolid at 5 cm depth, with height difference of 0.40 m in relation to the pressure switch, responsible for actuating each of the irrigation systems.

In each treatment, irrigation events were monitored, and SIC activation times were automatically recorded in a data collection and storage system composed of an Arduino Mega board and an SD card. Adapted from [24], to each irrigation event the system was programmed to interrupt the water supply for one minute every 30 seconds, preventing percolation in the plugs. If the power supply to the pressure switch remained active, the irrigation system was activated again. This procedure was repeated until the pressure switch was turned off by SIC, in response to the increase in water tension in the substrate.

The irrigation system was supplied by 12-V motor pumps (NeoSolar SingFlo DP-160), powered by two 130-W photovoltaic modules (YL140p-17b model, Yingu Solar) connected to a battery. For each type of water, there was an independent pumping system.

## Agroclimatic monitoring

Digital thermo-hygrometers (HT-4010 model, Icel) were installed inside the bench environments to determine relative humidity and temperature, with data storage interval of 30 minutes. Meteorological monitoring under the full sun condition was carried out by an automatic meteorological station, belonging to the National Institute of Meteorology (INMET) and located near the experimental site. In addition, rain collectors were installed under the conditions of full sun and inside the cover. With meteorological data, reference evapotranspiration (ETo) was estimated daily by the Penman-Monteith FAO-56 method [39].

## Seedling biometrics

Seedlings were evaluated from 01/06 to 10/09. Every 21 days, values of height (H) and collar diameter (D) of the seedlings were measured with graduated ruler and digital caliper, respectively. When about 50% of the seedlings acquired commercial standard of 30 cm in height and 3.0 mm in collar diameter [40, 41] three seedlings closer to the average of each repetition were selected to determine shoot dry matter (SDM), root system dry matter (RDM) and total dry matter (TDM). The selected seedlings were cut, separated into shoots and root system, placed in paper bags, and then taken to the oven at 65˚C and kept until reaching constant weight.

The collected data were used to calculate the relationship between height and collar diameter [41], also commonly known as H/D, and the Dickson quality index (DQI) [42].

$$DQI = \frac{TDM}{\left(\frac{H}{D} + \frac{SDM}{RDM}\right)} \tag{4}$$

where:

TDM is *total dry matter*, in g;

H is *height*, in cm;

D is *collar diameter*, in mm;

SDM is *shoot dry matter*, in g; and

RDM is *root dry matter*, in g.

Irrigation water productivity (WPir) (Eq 4) was calculated using the SDM, RDM and TDM as productivity (P), relative to the volume of water applied by irrigation (Va).

$$WPir = \frac{P}{Va} \tag{5}$$

where:

P is *productivity* in terms of total dry matter, in g; and

Va is the *total volume* of water applied by irrigation, in L.

## Planting seedlings in a forest restoration area

The seedlings not used for the previous analysis were hardened and planted in the Guapiaçu Ecological Reserve, inserted in Atlantic Forest biome (22˚27'32.26"S, 43˚45'53.72"W), which is an experimental area with 50 year of implantation, and where the forest restoration is in an advanced stage, with formed crowns and accumulated litter. The climate classification is of type Af, according to the Köppen, characterized as tropical with rainy summers and dry winters, without a markedly dry season, and average annual precipitation of 2,050 mm. The soil is classified as Cambisol Dystrophic Red Yellow Latosol. Automatic rain gauge from INMET located near the site recorded accumulated precipitation of 1,730 mm, from 2021 November to 2022 April.

The planting holes were dug between the rows, manually, with dimensions of 30 cm x 30 cm x 30 cm (length x width x depth), with a spacing of 2.5 m x 3.2 m between seedlings. The seedlings were planted on 10/22/2021, with six replications per treatment and their initial biometric data were recorded. The monitoring of the development of these plants was carried out at 150, 200 and 250 days after planting (DAP) for height and at 150 and 200 DAP for diameter.

## Statistical analysis

Analysis of variance (ANOVA) was performed and for that the normality and homogeneity of the residues were verified by the Shapiro-Wilk and Bartlett tests, respectively, at 5% probability.

The significance of the fit of the linear and quadratic models, of the biometric data height and diameter (H and D) was evaluated by the F test of the ANOVA, and their coefficients by the t-test, both with 95% probability level. The possible presence of trends in the estimates was validated by graphical analysis of relative residuals as a function of H and D. After evaluating the fitting statistics of the models, the one that showed the most satisfactory values in relation to the selection criteria described was selected [43].

With the regression model selected, the general fit was performed for each biometric variable, generating the sum of squares of the regression, the sum of squares of the residuals and the total sum of squares. In each stratification, the respective sums of squares were generated to make up the test of identity between models [44]. When there was no identity between models by F statistics at 95% probability level, it was concluded that it was not possible to use the same equation for different treatments, that is, they do not have identity [45].

For irrigation water productivity, Dickson quality index and nutrient extraction in nursery phase and the height and diameter of seedlings in a forest restoration area, Tukey test was performed at 5% probability level, for irrigation sources and within each cover. All analyses were performed with the help of computer programs R (3.6.0) and Sisvar (5.6).

# Results and discussion

## Quality of treated wastewater used in irrigation in nursery

After treatment by ozonation, the pH of the solution increased as the time of exposure to the gas increased, being 7.7 in the treatment of 1 h ($T_2$) and 7.8 in the treatment of 2 h ($T_3$) (Table 1).

**Table 1. Physicochemical characterization of wastewater in different treatment scenarios and for the control source.**

| Parameter | $T_1$* | $T_2$** | $T_3$** |
|---|---|---|---|
| | —————————concentrations————————— | | |
| pH | 6.4 | 7.6 (0.23) | 7.8 (0.30) |
| C.E (dS m$^{-1}$) | 0.96 | 1.76 (0.96) | 1.64 (1.18) |
| ST | 9.8 | 360 (91.49) | 246.50 (94.22) |
| Turbidity (NTU) | 1.83 | 105.80 (38.25) | 75 (48.14) |
| Ca$^{2+}$ (mg L$^{-1}$) | 2.03 | 18.23 (7.29) *** | 17.84 (6.74) *** |
| TOC | <1.0 | 4.27 (2.18) | 3.07 (1.93) |
| $P_t$ (mg L$^{-1}$) | 0.24 | 15.47 (11.57) | 11.64 (6.81) |
| Mg$^{2+}$ Total (mg L$^{-1}$) | 0.71 | 12.66 (3.57) *** | 13.08 (3.46) *** |
| NH$_4^+$ (mg L$^{-1}$) | 0.1 | 26.76 (10.48) | 18.82 (2.35) |
| Na$^+$ (mg L$^{-1}$) | 2.36 | 29.10 (5.15) *** | 32.84 (6.48) *** |
| SAR | 2.02 | 7.48 (0.15) *** | 8.41 (0.47) *** |

$T_1$: control water; $T_2$: ozonated cattle wastewater for 1 h; $T_3$: ozonated cattle wastewater for 2 h

*values provided by the supply company

**mean of the concentrations, n = 5 and

*** n = 3; SD in parentheses next to the means.

The oxidative process of ozonation occurs in two distinct ways: direct electrolytic attack by molecular ozone and indirect attack by OH• radicals produced by the decomposition of $O_3$ [11, 46]. The reaction that occurs from the contact of the ozone gas with the organic compounds of the effluent releases hydroxide ions, making them dominant over $H^+$ ions and consequently resulting in a more basic medium. Used for irrigation, the CWW treatments do not cause risk of soil acidification, remaining above neutrality.

There was a decrease of 30 to 35% in electrical conductivity after ozonation. According to the values of $Ca^+$ and $Mg^{2+}$, it is possible to note that the lower exposure to ozonation ($T_2$) preserved these cations, leading to higher concentrations compared to the treatment of higher exposure to $O_3$ ($T_3$). Conversely, $Na^+$ was favored by the longer ozonation time. When observing the relationship of these exchangeable cations through the sodium adsorption ratio (SAR), values of 7.4 and 8.35 were found for $T_2$ and $T_3$, respectively. Based on the EC and SAR values recommended by [47], the ozonation treatments for 1 h and 2 h have potential for use in irrigation, causing no reduction in substrate infiltration capacity.

CWW contained an average of 969.8 mg $L^{-1}$ of total solids in its composition, and exposure to ozone drastically reduced this value, reaching 360.0 and 246.5 mg $L^{-1}$ in $T_2$ and $T_3$, respectively. In addition to the upward drag of particles by the gas, which were trapped in the cotton waste rag that closed the system superficially, there was destruction of solid particles of organic matter, especially at the beginning of the reaction [48]. This fact clarifies that there are no major differences between the results of ozonation treatments.

Turbidity decreased by more than 50% with ozonation for 1 h and by more than 65% with ozonation for 2 h. The concentrations are similar to those found by [12], who recommended the use of treatments for drip irrigation, since TSS values were close to 50 mg $L^{-1}$ [49], causing no clogging of emitters. Indeed, during the entire experimental period, no clogged emitters were identified.

The lower exposure to ozonation promoted higher concentrations of P, N and OM, which constitute interesting results for agricultural use of $O_3$ as an enhancer of substrate fertility. The use of $O_3$ is also a well-known method for ammonia oxidation [12], and the reaction of $O_3$ with ammonia can be expressed as in Eq 5. This explains the reduction of ammonia by approximately 45 and 62% in $T_2$ and $T_3$, respectively.

$$NH_4^+ + 4O_3 \rightarrow 2H^+ + NO_3^- + H_2O + 4O_2 \tag{6}$$

## Meteorologicaaspects and volume applied by irrigation in nursery

The accumulated reference evapotranspiration (ETo Ac) was 300.4 mm (Fig 2), with a maximum value of 34.5 mm in the 12th week. As the experiment was conducted during the winter period, the least rainy season in the region, there were only 24 days with precipitation, which totaled 95.8 mm (Ppt Ac). In the 110 days of experiment, precipitation was higher than ETo in just 13 days, surpassing the reference evapotranspiration only in the second week of the experiment.

In general, there was a higher frequency of actuation of the irrigation system under the full sun condition ($C_1$) (Table 2), where the weather conditions favor greater evaporation of water from the surface of the tube. As the porous cup of the controller reflects the moisture conditions in the substrate, it becomes drier and, consequently, the signal is sent to the pressure switch more frequently. However, the mean irrigation time was lower, except for the $T_3$ treatment. Under shading levels ranging from 0% to 92% and using the same water as the $T_1$ treatment, evaluated the growth of *D. nigra*, for 115 days (July to October) in the same experimental area [24]. For an accumulated ETo of 436.5 mm, the irrigation system was activated 49 times on average, with an average activation time of 2.8 min.

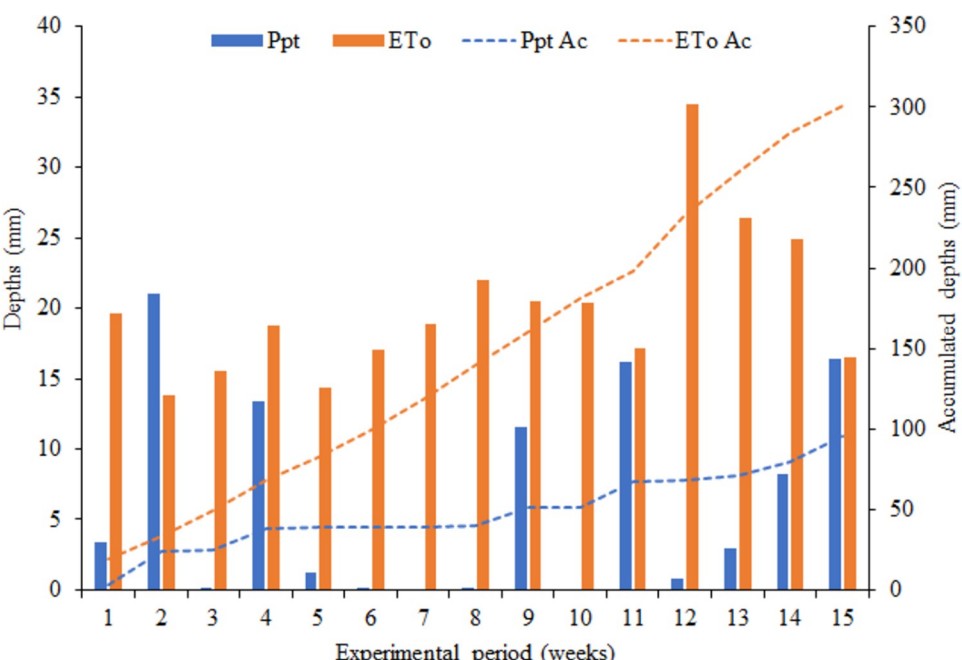

**Fig 2. Precipitation (Ppt) and reference evapotranspiration (ETo) depths along the experimental period.**

The growth rate of the plant also influences the water requirement, causing the number of actuations of the system for plants irrigated with municipal supply water ($T_1$) and cultivated under shading ($C_2$) to be equal to that for the full sun condition. Under this condition, there was a higher water requirement by plants, compared to the volume applied to plants irrigated with CWW treated (Fig 3B).

In all treatments, there were days with more than one actuation in response to the water need of the plants, which occurred, on average, from the 7th experimental week. When municipal supply water was used ($T_1$), there were up to 3 actuations per day in the 13th and 14th weeks (Fig 3), and the same occurred with the water from the $T_2$ treatment, under the full sun condition. For the $T_3$ treatment, the number of actuations was lower and irrigation times were shorter.

The average volumes of water applied by the system varied according to the water need of the plants and the characteristic of the irrigation water. Under the full sun condition ($C_1$), the accumulated volumes were 1.653, 2.096 and 1.348 L plant$^{-1}$, for treatments $T_1$, $T_2$ and $T_3$, respectively (Fig 3A), while under the cover $C_2$, the volumes were 1.921, 1.142 and 1.171 L

**Table 2. Actuation of irrigation systems to produce *D. nigra* seedlings under different cover conditions and types of water.**

| Cover | Types of water | Actuation day | System actuation | Average irrigation time (s) |
|---|---|---|---|---|
| $C_1$ | $T_1$ | 50 | 61 | 38 |
| | $T_2$ | 57 | 72 | 41 |
| | $T_3$ | 38 | 41 | 39 |
| $C_2$ | $T_1$ | 43 | 61 | 50 |
| | $T_2$ | 27 | 40 | 46 |
| | $T_3$ | 35 | 38 | 36 |

$T_1$ –control water; $T_2$—ozonated cattle wastewater for 1 h; $T_3$—ozonated cattle wastewater for 2 h

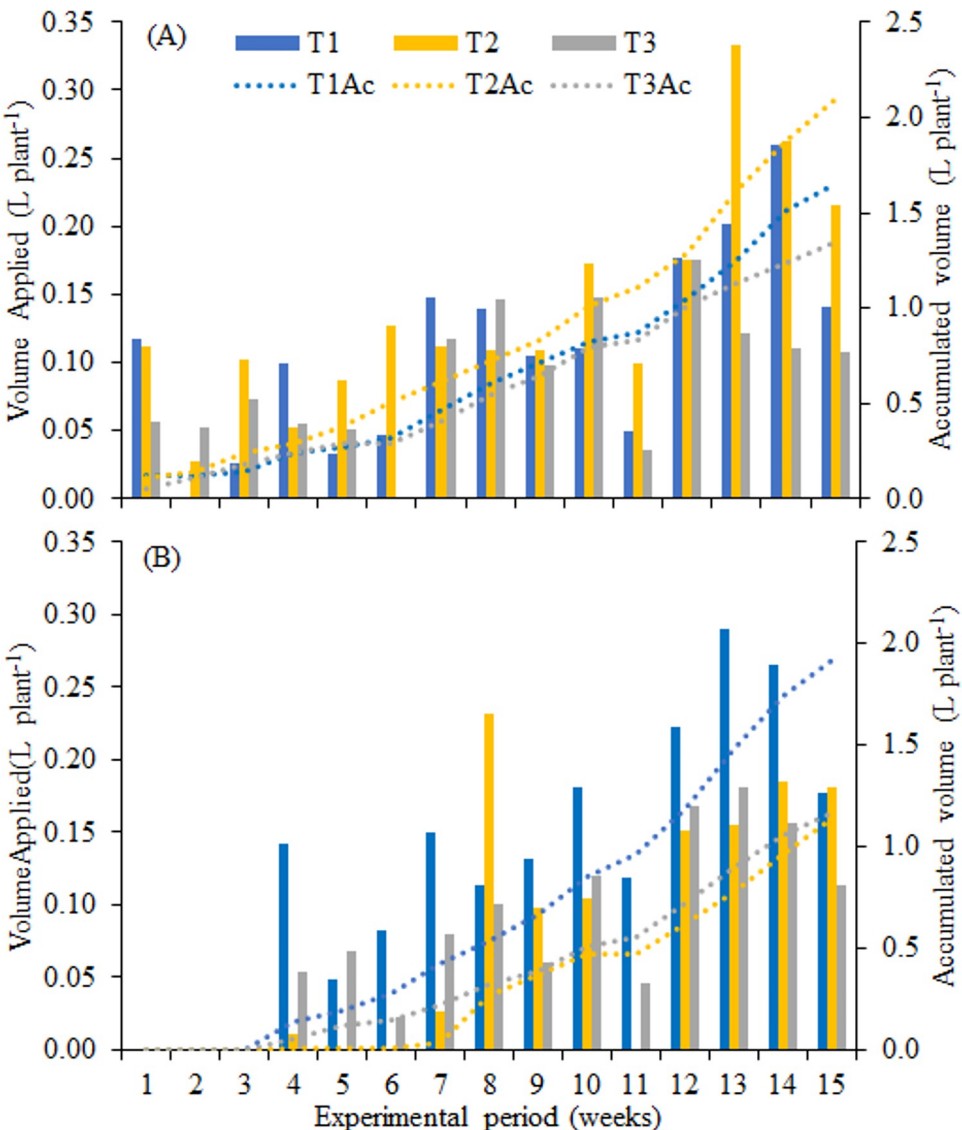

**Fig 3.** Volume of water applied per plant under conditions of full sun (A) and with cover (B), for the different treatments. $T_1$ –control water; $T_2$ -ozonated cattle wastewater for 1 h; $T_3$—ozonated cattle wastewater for 2 h.

plant$^{-1}$ (Fig 3B). Plants irrigated with municipal supply water ($T_1$) received higher volume of water when cultivated under the cover $C_2$, but when irrigated with CWW treated there was higher requirement under the full sun condition. The total applied by [24] was, on average, 2,807 L per seedling of *D. nigra*.

The amount of water required by plants depends both on local weather conditions and on their growth stage. Under the cover $C_2$, irrigation systems were not actuated during the first three weeks of the experiment (Fig 3B). In addition to precipitation (24.6 mm), the plants were at the beginning of the growth stage and the microclimate formed by the cover promoted less evaporation, favoring greater water retention in the substrate.

In full sun cultivation (Fig 3A), the water requirements of plants irrigated with water from the treatments $T_1$ and $T_3$ were similar until the 12th week, while under the cover $C_2$ the requirement was similar from the 8th week for plants irrigated with CWW treated and, on

average, 40% lower than the requirement of plants irrigated with municipal supply water. This fact can be explained by the formation of a biofilm on the tube's surface, caused by the higher organic load of the wastewater applied in irrigation associated with the microclimate created by the cover. The presence of the primarily photosynthetic microorganisms can provide benefits to the substrate, such as the improvement of soil structure [50]. The authors state that, when formed in the soil, algae biomass can improve physical properties such as water retention, acting as a potential soil conditioner.

Cementation occurs from cellular debris and excretions of extracellular polysaccharides of microbial origin. Such cementation causes the formation of an evident biofilm on the soil surface, affecting infiltration, percolation, water retention etc. [51]. In general, biofilm acts as a buffering of the system, delaying the action of meteorological agents in the vertical section of the tube and increasing water use efficiency.

## Growth and quality of forest seedlings in nursery

*D. nigra* seedlings were collected at 110 days after emergence (DAE), when most of them showed shoot height and collar diameter according to standards to be taken to the field, according to [41]. Mortality rates were lower than 10% and are not associated with a specific treatment. In full sun, *D. nigra* seedlings showed increasing variation of height, following a linear trend (Fig 4A), regardless of the type of irrigation water. Using $T_2$, the plants showed higher height growth in relation to the others, with a rate of 0.3339 cm day$^{-1}$. Under the cover condition, the result was similar for $T_1$, but for $T_2$ and $T_3$ the growth followed a second-order polynomial trend (Fig 4C). For collar diameter, there was a second-order polynomial trend (Fig 4B) in all treatments, except for $T_2$ under the cover $C_1$ (Fig 4D), which showed a linear fit. The growth in height and diameter for both covers showed no identity (p <0.05) between treatments. The results for T1 and T3 are similar to those obtained by [24], considering

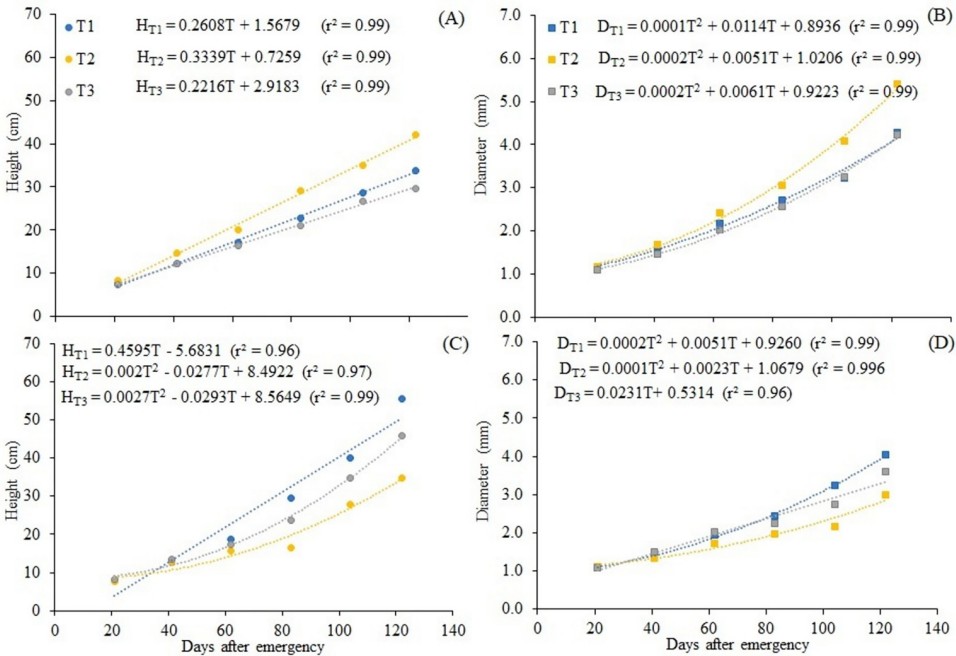

**Fig 4.** Growth of *D. nigra* seedlings in height and diameter as a function of the days after emergence (DAE), produced under full sun (A and B) and 50% cover (C and D). (*p<0.05).

shading levels from 0% to 58%. However, the plants reached a diameter greater than 5.0 mm when irrigated with $T_2$, showing the potential for using treated CWW.

The seedlings from the $C_2$ cover reached the minimum height for field planting (30 cm) at about 75 DAE for $T_1$, 110 DAE for $T_2$ and 92 DAE for $T_3$, while in the full sun this height was reached at 108 DAE ($T_1$), 88 DAE ($T_2$) and 120 DAE ($T_3$). The recommended diameter for planting (3.0 mm) was reached first under full sun, at 92, 80 and 96 DAE, for $T_1$, $T_2$ and $T_3$, respectively, and later under Sombrite$^®$ cover, at 96, 120 and 105 DAE, respectively. The commercial standard was reached faster in $T_2$ under full sun and in $T_1$ and $T_3$ under the cover condition. Producing quality seedlings in less time is seen as an advantage to maximize the use of nurseries [24].

Despite the differences between the models presented in Fig 4, the final height and diameter differed between the treatments evaluated (Table 3). Plants irrigated with $T_2$ showed higher values (42.03 cm and 5.40 mm) under full sun. With shading, plants irrigated with $T_1$ showed higher values of height and diameter. Following the same trend, plants irrigated with $T_1$ and $T_2$ showed higher DQI under the conditions $C_2$ and $C_1$, respectively. Indices such as the DQI, which combine the allometric variables of the plant, are used and proven to serve as predictors for the establishment of seedling development [52]; the higher the value of this index, the better the quality of the seedling produced.

Under the cover $C_2$, biofilm formed on the surface of the tubes, which contributed to lower application of water to the plants, resulting in the lower supply of nutrients present in the CWW treated. When water from the $T_2$ treatment was used, plants showed the lowest mean values of height (34.75 cm) and diameter (2.99 mm) (Table 3).

The standards achieved under shading by the seedlings of treatments $T_1$ and $T_3$ did not differ significantly, but the treatment with ozonated wastewater once again stood out with regard to water saving and irrigation water productivity (Table 3). The WPir found in the $T_2$ treatment were 286% and 30.8% higher than those obtained by [24], for $C_1$ and $C_2$, respectively.

The growth of *D. nigra* seedlings in the different treatments and covers showed identity in its models, that is, the behavior is similar between water treatments, for each cover (Fig 5). This is another way of highlighting the production efficiency of $T_2$ under full sun, making use of water and nutrient resources, to achieve greater height and diameter in a shorter time.

Under the cover $C_2$, it is evident that the planting standards of the seedlings were achieved with the supply of approximately 1200 mL per plant, regardless of the water used in irrigation. Under full sun and with the same amount of water, only the seedlings produced with $T_1$ and $T_2$ reached this standard.

**Table 3. Morphological attributes of *D. nigra* seedlings under full sun ($C_1$) and under cover with 50% shading ($C_2$) at 122 days after emergence (DAE).**

| Cover | Types of water | $H_{END}$ (cm) | $D_{END}$ (mm) | DQI | WPir (g L$^{-1}$) |
|---|---|---|---|---|---|
| $C_1$ | $T_1$ | 33.85b | 4.28b | 0.26b | 1.57b |
|  | $T_2$ | 42.03a | 5.40a | 0.47a | 2.35a |
|  | $T_3$ | 29.03b | 4.22b | 0.25b | 1.71b |
| $C_2$ | $T_1$ | 55.84a | 3.95a | 0.17a | 1.48b |
|  | $T_2$ | 34.75c | 2.99b | 0.10b | 1.24a |
|  | $T_3$ | 45.73b | 3.60a | 0.15ab | 2.01a |

$T_1$: control water; $T_2$: ozonated cattle wastewater for 1 h; $T_3$: ozonated cattle wastewater for 2 h; $H_{END}$: final height; $D_{END}$: final diameter; DQI: Dickson quality index; WPir: irrigation water productivity in relation to total dry matter. Means followed by the same letter in the column, for each cover, do not differ statistically by Tukey test at 5% probability level. (n = 12).

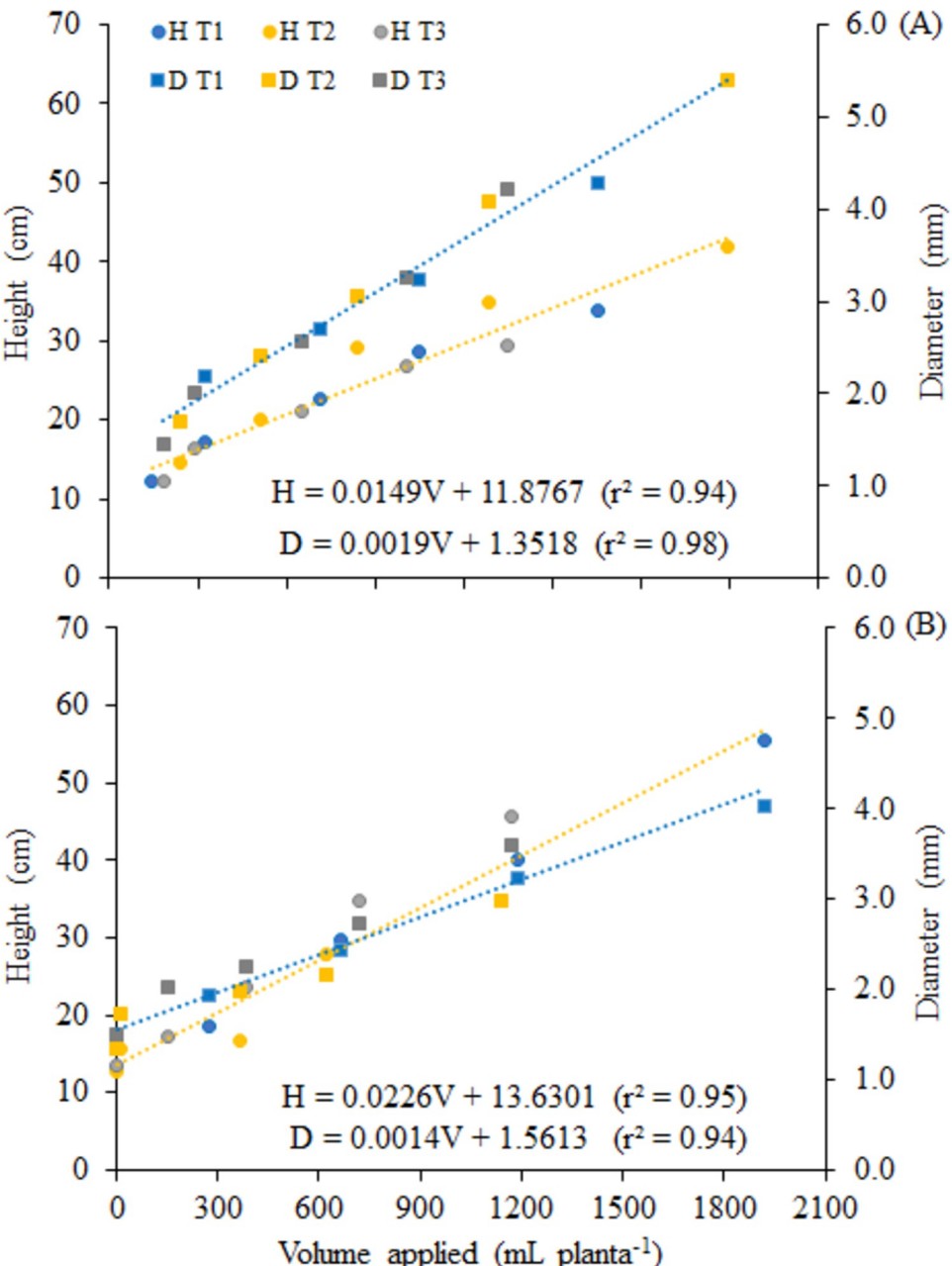

**Fig 5.** Growth of *D. nigra* seedlings in height and diameter as a function of the volume irrigated per plant, produced under full sun (A) and under shading (B) (*p<0.05).

### Growth and initial quality of seedlings in a forest restoration area

The growth of *D. nigra* seedlings in height and diameter up to 200 DAP is shown in Fig 6. After the hardening period, seedlings produced under full sun ($C_1$) were planted with statistically similar heights (Fig 6A), but seedlings produced with control water ($T_1$) showed lower growth after 150 DAP. For diameter (Fig 6B), the seedlings produced in $T_1$ already showed inferiority at planting, not reaching the same level as $T_2$ and $T_3$ in any of the evaluations.

The seedlings produced in the shading condition ($C_2$) did not show statistical differences in the parameters evaluated during the sampled period (Fig 6C and 6D), however the seedlings

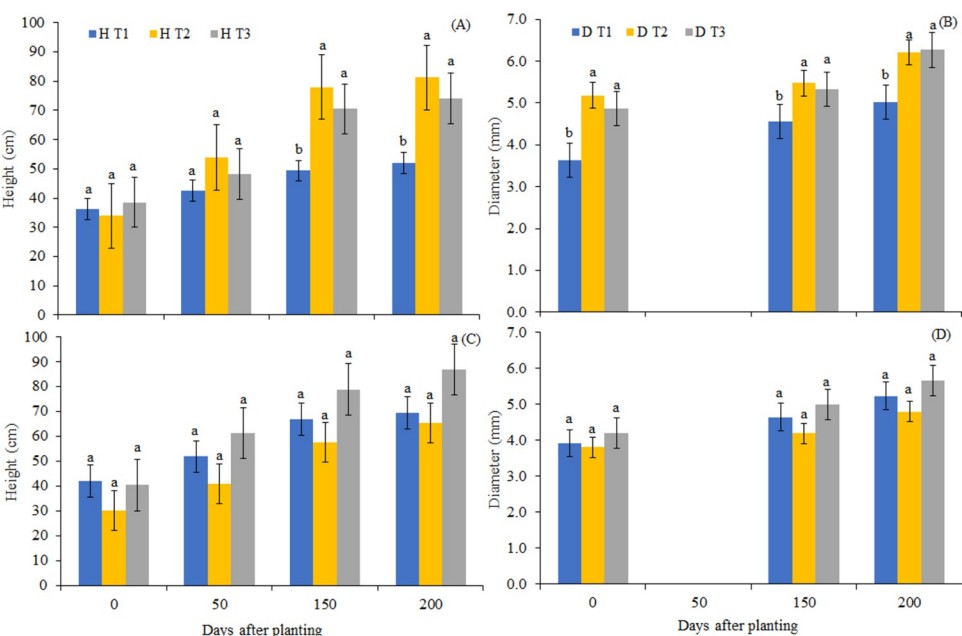

**Fig 6.** Growth in height and diameter of seedlings of *D. nigra*, planted in the forest restoration area of the Atlantic Forest, under $C_1$ (A and C) and $C_2$ (B and D) covers. $C_1$: full sun; $C_2$: 49.4% shading; T1: control water; T2: Ozonation for 1h; T3: Ozonation for 2h. Means followed in the column by the same letter do not differ from each other by Tukey's test, at 5% probability; (n = 6).

produced with CWW treated ($T_2$ and $T_3$) had a height increase of the order 116 .1% and 114.8%, respectively, indicating a future differentiation in relation to seedlings produced with control water ($T_1$).

Although the evaluation period is short for monitoring forest species, the results obtained at 200 DAP indicate benefits when producing seedlings irrigated with treated CWW, since there was greater nutrients supply in the nursery stage and gradually absorb the nutrients remaining in the substrate. These results corroborate with [53], who evaluated the effect of wastewater irrigation on nectarine physiology. The authors concluded that treated wastewater positively affected vegetative growth, especially in terms of shoot length. For forest species, no follow-up studies of seedlings produced with wastewater were found.

## Conclusions

In this study, we evaluated the quality of cattle wastewater (CWW) treated by ozonation, the water requirement and its effect on the growth of seedlings of *Dalbergia nigra* cultivated with sewage sludge under different light conditions. In addition, the height and diameter of seedlings transplanted in a forest restauration area (FRA) of Atlantic Forest were evaluated for 200 days. Irrigation with cattle wastewater (CWW) treated by ozonation did not compromise the development of *D. nigra* seedlings. Under conditions of full sun, CWW ozonated for 1 h promoted a better standard of dispatch of *D. nigra* seedlings, reducing the production time by 20 days considering the same microclimate. Furthermore, plants irrigated with T2 had higher DQI (0.47) and WPir (2.35 g $L^{-1}$). When cultivated under shading of approximately 50%, *D. nigra* seedlings reach shorter nursery time when irrigated with municipal supply water but using T3 the plants showed higher WPir (2.01 g $L^{-1}$). The initial vegetative growth of the seedlings planted in RFA was benefited by the nutrients provided by the CWW treated. It is possible to produce forest seedlings with sewage sludge and cattle wastewater, which represent

actions aimed at converting waste into nutrients and disposing of by-products, actions that are encompassed in the concept of circular economy.

## Supporting information

**S1 Dataset.**
(RAR)

## Acknowledgments

We thank the Federal Rural University of Rio de Janeiro, specifically the PPGACS, GPASSA, GPMI and LAPER.

## Author Contributions

**Conceptualization:** Laiz de Oliveira Silva, Henrique Vieira Mendonça, Daniel Fonseca de Carvalho.

**Data curation:** Laiz de Oliveira Silva, Henrique Vieira Mendonça, Marinaldo Ferreira Pinto, Daniel Fonseca de Carvalho.

**Funding acquisition:** Daniel Fonseca de Carvalho.

**Investigation:** Laiz de Oliveira Silva, Bruno Antonio Augusto Faria Conforto.

**Methodology:** Laiz de Oliveira Silva, Henrique Vieira Mendonça, Marinaldo Ferreira Pinto, Daniel Fonseca de Carvalho.

**Writing – original draft:** Laiz de Oliveira Silva, Henrique Vieira Mendonça, Daniel Fonseca de Carvalho.

**Writing – review & editing:** Laiz de Oliveira Silva, Henrique Vieira Mendonça, Daniel Fonseca de Carvalho.

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
