## [Decision Letter · Decision Letter 0]

22 Aug 2022

PONE-D-22-17230PRODUCTION OF FOREST SEEDLINGS USING SEWAGE SLUDGE AND OZONATED CATTLE WASTEWATER: A POTENTIAL FOR CIRCULAR ECONOMYPLOS ONE

Dear Dr. Carvalho,

Thank you for submitting your manuscript to PLOS ONE. After careful consideration, we feel that it has merit but does not fully meet PLOS ONE’s publication criteria as it currently stands. Therefore, we invite you to submit a revised version of the manuscript that addresses the points raised during the review process.

We look forward to receiving your revised manuscript.

Kind regards,

Shijian Ge, Ph.D.

Academic Editor

PLOS ONE

2. In your Methods section, please provide additional information regarding the permits you obtained to collect samples for the present study. Please ensure you have included the full name of the authority that approved the field site access and, if no permits were required, a brief statement explaining why.

4. Please include a copy of Table 2 which you refer to in your text on page 14.

Reviewers' comments:

Reviewer's Responses to Questions

**Comments to the Author**

1. Is the manuscript technically sound, and do the data support the conclusions?

Reviewer #1: No

Reviewer #2: Partly

2. Has the statistical analysis been performed appropriately and rigorously? 

Reviewer #1: N/A

Reviewer #2: No

3. Have the authors made all data underlying the findings in their manuscript fully available?

Reviewer #1: Yes

Reviewer #2: Yes

4. Is the manuscript presented in an intelligible fashion and written in standard English?

Reviewer #1: No

Reviewer #2: No

5. Review Comments to the Author

Reviewer #1: 1. References are not well cited. Kindly update those; for example in line 80.

2. Introduction is not clear, i hope that they will make investigate the current methods' advantages disadvantages and finally propose their research work. Line 110-112, you have proposed your work and why its necessary?

3. Kindly make a comparison of currently available methods and compare your research results with them.

Reviewer #2: The study represents the application of sewage sludge and cattle wastewater (CWW) to produce forest seedlings, attempting to reduce the release of waste and effluents into the environment. Generally, the results/findings are fine but some points should be considered before acceptance:

1- The title includes the expression “Circular Economy”; however, it disappeared throughout the manuscript!! Why?

2- Do not repeat the words in the title to the “keywords”.

3- In the Introduction part, give some examples from the literature for the application of sewage/waste for forest seedlings.

4- The last paragraph of Introduction should include the study objectives/procedures in brief.

5- The study should unveil the major gaps within the existing knowledge of the sewage/waste utilization for forest seedlings.

6- Are there any results for control run, without using sewage/waste?

7- The results should be enriched with statistical explanations.

8- The abstract and conclusion sections should be improved to show the main research findings.

6. PLOS authors have the option to publish the peer review history of their article (what does this mean?). If published, this will include your full peer review and any attached files.

Reviewer #1: No

Reviewer #2: No

---

## [Author Response · Author response to Decision Letter 0]

13 Sep 2022

To the Editorial Board of PLOS ONE 

Dear Editor,

We would like to thank you for your message and for the reviewer’s comments on our manuscript (PONE-D-22-17230), entitled “Production of forest seedlings using sewage sludge and ozonated cattle wastewater: a potential for circular economy”. 

We have modified the manuscript in response to the relevant and constructive reviewer comments. We highlight the modifications referring to the comments of the Reviewer 1 in red, and the Reviewer 2 in blue, and in green we present some modifications that, although they were not mentioned by the reviewers, also contributed to the improvement of the text, according to our judgment. It is worth mentioning that due to one of the comments by Reviewer #2, the title of the article was changed. We hope the revised version is now suitable for publication, however we are available for any clarifications. 

In the sequence we present the Responses to each point raised by the academic editor and reviewer.

Best regards,

Daniel Fonseca de Carvalho (Corresponding author)

Journal Requirements

Response: all the format, style and names are checked.

2. In your Methods section, please provide additional information regarding the permits you obtained to collect samples for the present study. Please ensure you have included the full name of the authority that approved the field site access and, if no permits were required, a brief statement explaining why.

Response: Thanks for the comment. The experiment was carried out at a public university, where some authors are professors and others are undergraduate and graduate students. Therefore, there was no need for formal authorization to carry out this study. This information was included in lines 128-129.

Response: The main data collected in the field experiment were made available in the journal's system.

4. Please include a copy of Table 2 which you refer to in your text on page 14.

Response: Indeed, at the time of submission, the numbering of the tables was changed due to text links. This has been fixed in the new version.

Responses to the reviewer’s comments

Reviewer #1

1. References are not well cited. Kindly update those; for example in line 80.

Response: All references have been checked and adapted to the journal's rules. Thanks.

2. Introduction is not clear, i hope that they will make investigate the current methods' advantages disadvantages and finally propose their research work. Line 110-112, you have proposed your work and why its necessary?

Response: Changes have been made to the text in the paragraph citing the benefits of ozonation (lines 82-29).

3. Kindly make a comparison of currently available methods and compare your research results with them.

Response: In lines 79-81, we have mentioned the main methodologies for secondary treatment using advanced oxidation processes, recommended for organic effluents. However, in this study we used the ozonation methodology and no comparisons were made between CWW treatment methodologies.

Reviewer #2: The study represents the application of sewage sludge and cattle wastewater (CWW) to produce forest seedlings, attempting to reduce the release of waste and effluents into the environment. Generally, the results/findings are fine but some points should be considered before acceptance:

1- The title includes the expression “Circular Economy”; however, it disappeared throughout the manuscript!! Why?

Response: You are right: we included this expression because it represents the logic of our study, by reusing waste and cycling nutrients that would be lost and potentially polluting if disposed of incorrectly. However, although our technique fits the principles of the circular economy, we did not emphasize this in the text and chose to change the title.

2- Do not repeat the words in the title to the “keywords”.

Response: with the new title, the keywords were changed and we chose to add the term "circular economy".

3- In the Introduction part, give some examples from the literature for the application of sewage/waste for forest seedlings. 

Response: We did not find in the literature studies that used cattle wastewater in the production of forest seedlings. However, studies carried out with sewage sludge were cited in lines 97 to 99.

4- The last paragraph of Introduction should include the study objectives/procedures in brief.

Response: The text referring to the objective of the study is presented in lines 117 to 119.

5- The study should unveil the major gaps within the existing knowledge of the sewage/waste utilization for forest seedlings.

Response: As we commented in one of the answers, the use of sewage sludge for the production of seedlings of tree species has increased in Brazil, but in none of the studies that have already been carried out has been used cattle water. Thus, we understand that this is a great contribution of our article.

6- Are there any results for control run, without using sewage/waste?

Response: thanks for your comment. Research groups from our University, in the forestry area, have been working with sewage sludge for many years, but without quantifying the volume of water applied in the production of forest seedlings. We started studying the water requirements of seedlings in 2018, maintaining the theme of using sewage sludge as a substrate for seedlings. Therefore, we did not intend to evaluate the growth of seedlings in another type of substrate. We intend to make this comparison in the next experiments. Thanks.

7- The results should be enriched with statistical explanations.

Response: Although no results were found for the production of seedlings with CWW, the results of the study were compared with those obtained using water from the T1 treatment (lines 339-343; 368-369; 396-397; 401-404; 437-439).

8- The abstract and conclusion sections should be improved to show the main research findings.

Response: the text has been changed.

---

## [Decision Letter · Decision Letter 1]

11 Oct 2022

Production of forest seedlings using sewage sludge and automated irrigation with ozonated cattle wastewater

PONE-D-22-17230R1

Dear Dr. Daniel Fonseca de Carvalho,

We’re pleased to inform you that your manuscript has been judged scientifically suitable for publication and will be formally accepted for publication once it meets all outstanding technical requirements.

Kind regards,

Shijian Ge, Ph.D.

Academic Editor

PLOS ONE

Additional Editor Comments (optional):

Reviewers' comments:

Reviewer's Responses to Questions

**Comments to the Author**

1. If the authors have adequately addressed your comments raised in a previous round of review and you feel that this manuscript is now acceptable for publication, you may indicate that here to bypass the “Comments to the Author” section, enter your conflict of interest statement in the “Confidential to Editor” section, and submit your "Accept" recommendation.

Reviewer #1: (No Response)

Reviewer #2: All comments have been addressed

2. Is the manuscript technically sound, and do the data support the conclusions?

Reviewer #1: (No Response)

Reviewer #2: Partly

3. Has the statistical analysis been performed appropriately and rigorously? 

Reviewer #1: (No Response)

Reviewer #2: Yes

4. Have the authors made all data underlying the findings in their manuscript fully available?

Reviewer #1: (No Response)

Reviewer #2: Yes

5. Is the manuscript presented in an intelligible fashion and written in standard English?

Reviewer #1: (No Response)

Reviewer #2: Yes

6. Review Comments to the Author

Reviewer #1: (No Response)

Reviewer #2: The authors' responses are satisfactory. Essential experiment would be used as a control run, without using sewage/waste

7. PLOS authors have the option to publish the peer review history of their article (what does this mean?). If published, this will include your full peer review and any attached files.

Reviewer #1: No

Reviewer #2: **Yes: **Assoc. Prof. Mahmoud Nasr

---

## [Editor Report · Acceptance letter]

21 Oct 2022

PONE-D-22-17230R1 

Production of forest seedlings using sewage sludge and automated irrigation with ozonated cattle wastewater 

Dear Dr. Carvalho:

I'm pleased to inform you that your manuscript has been deemed suitable for publication in PLOS ONE. Congratulations! Your manuscript is now with our production department. 

Kind regards, 

on behalf of

Professor Shijian Ge 

Academic Editor

PLOS ONE